

# Comparative transcriptome analysis during seeds development between two soybean cultivars

Li Peng[1],*, Linlin Qian[1],*, Meinan Wang[1], Wei Liu[1], Xiangting Song[1], Hao Cheng[1], Fengjie Yuan[2] and Man Zhao[1]

[1] College of Bioengineering and Biotechnology, Zhejiang University of Technology, Hang Zhou, China
[2] Institute of Crop Science, Zhejiang Academy of Agricultural Sciences, Hang Zhou, China
* These authors contributed equally to this work.

## ABSTRACT

Soybean is one of the important economic crops, which supplies a great deal of vegetable oil and proteins for human. The content of nutrients in different soybean seeds is different, which is related to the expression of multiple genes, but the mechanisms are complicated and still largely uncertain. In this study, to reveal the possible causes of the nutrients difference in soybeans A7 (containing low oil and high protein) and A35 (containing high oil and low protein), RNA-seq technology was performed to compare and identify the potential differential expressed genes (DEGs) at different seed developmental stages. The results showed that DEGs mainly presented at the early stages of seeds development and more DEGs were up-regulated at the early stage than the late stages. Gene Ontology and Kyoto Encyclopedia of Genes and Genomes analysis showed that the DEGs have diverged in A7 and A35. In A7, the DEGs were mainly involved in cell cycle and stresses, while in A35 were the fatty acids and sugar metabolism. Specifically, when the DEGs contributing to oil and protein metabolic pathways were analyzed, the differences between A7 and A35 mainly presented in fatty acids metabolism and seeds storage proteins (SSPs) synthesis. Furthermore, the enzymes, fatty acid dehydrogenase 2, 3-ketoacyl-CoA synthase and 9S-lipoxygenase, in the synthesis and elongation pathways of fatty acids, were revealed probably to be involved in the oil content difference between A7 and A35, the SSPs content might be due to the transcription factors: *Leafy Cotyledon 2* and Abscisic *acid-intensitive 3*, while the sugar transporter, SWEET10a, might contribute to both oil and protein content differences. Finally, six DEGs were selected to analyze their expression using qRT-PCR, and the results were consistent with the RNA-seq results. Generally, the study provided a comprehensive and dynamic expression trends for the seed development processes, and uncovered the potential DEGs for the differences of oil in A7 and A35.

Corresponding author
Man Zhao, mzhao@zjut.edu.cn

## INTRODUCTION

Soybean (*Glycine max* L. Merr) is a staple economic crop, which provides about one-third of the protein and vegetable oil for human diet (*Graham & Vance, 2003*). The contents of oil and protein vary among soybean varieties, with seed oil and protein ranges from 15% to 25% and 35% to 50%, respectively (*Hurburgh, 1994*). Soybean oil is mainly composed of unsaturated oleic, linoleic and linolenic acid, and their proportion determines the quality of the oil. Seeds storage proteins (SSPs) in soybean, the other important nutrient, are composed of β-conglycinin (7S) and glycinin (11S) (*Harada, Barker & Goldberg, 1989*; *Nielsen et al., 1989*; *Meinke, Chen & Beachy, 1981*). In soybean breeding, one of the important objects is to improve the contents of oil and protein. However, owing to the significantly negative correlation between oil and protein contents, it is difficult to develop soybean lines with high content of both nutrients (*Hyten et al., 2004a*, *2004b*; *Li et al., 2018a*, *2018b*; *Wilcox & Shibles, 2001*).

In plants, seed development is a dynamic process, which is regulated by both genetic and environmental factors. The accumulation of the nutrients oil, carbohydrate and protein, is regulated by the programmed expression of a metabolic network during seed development (*Gupta et al., 2017*; *Le et al., 2007*). Numerous studies have been performed to reveal the genetic basis of seed development (*Akond et al., 2014*; *Lee, Bilycu & Shannon, 2007*; *Hwang et al., 2014*; *Hills, 2004*). For example, the studies of quantitative trait loci (QTL) and genome-wide association studies (GWAS) have identified more than 100 QTLs related to the oil and protein contents, which are widely distributed over the 20 soybean chromosomes (*Hyten et al., 2004a*, *2004b*; *Akond et al., 2014*; *Lee, Bilycu & Shannon, 2007*; *Korte & Farlow, 2013*; *Panthee, Pantalone & Saxton, 2006*; *Phansak et al., 2016*; *Wang et al., 2014*). Furthermore, the sugar transporter GmSWEET10a which conferred simultaneous increases in soybean seed size and oil content as well as reduction in protein content, was identified through resequencing 800 genotypes (*Wang et al., 2020*).

The final chemical seed composition is a consequence of gene expression during seed development. Along with the development in "omics"-areas, the genes and metabolites that are required for seed development have been studied systematically (*Li et al., 2015*). Analyses of gene expression during seed development have provided clues for understanding final seed composition in different species, for example, *Arabidopsis thaliana* (*Ruuska et al., 2002*; *Palovaara, Saiga & Weijers, 2013*; *Fait et al., 2006*), *Medicago truncatula* (*Fedorova et al., 2002*; *Gallardo et al., 2007*), *Brassica napus* (*Li et al., 2005*), rice (*Lan et al., 2004*; *Furutani, Sukegawa & Kyozuka, 2006*), barley (*Watson & Henry, 2005*) and soybean (*Collakova et al., 2013*). In soybeans, transcriptome analysis of different developmental stages of seeds revealed that the most abundantly expressed genes contributing to the metabolism and accumulation of oil and SSPs mainly occurred at the middle and late stages (*Li et al., 2015*; *Libault et al., 2010*; *Severin et al., 2010*). For example, during the synthesis of SSPs, the 7S and 11S SSPs were highly expressed at the late stage, a process regulated by many transcription factors (*Severin et al., 2010*; *Verdier & Thompson, 2008*). On the other hand, many synthesis genes of amino acids and genes encoding proline-rich proteins were highly expressed at the early seed development stages.

As for lipids, there was a programmed expression of lipid biosynthesis-related genes, in which FAD2-2B and FAD2-2C were highly expressed at early stages and FAD2-1A and FAD2-1B were highly expressed at later stages (*O'Rourke et al., 2014*). GWAS analysis revealed that Glyma.11G229600.1 was positively and Glyma.04G102900.1 was negatively correlated with the oleic acid content in soybeans (*Liu et al., 2020*).

In this study, RNA-Seq technology was applied to study three stages of seed development in two different soybean cultivars with distinct protein and oil content. Through comparative transcriptome analysis of developing seeds, we tried to reveal dynamic gene expression trends, and identify the important differential expressed genes (DEGs) and metabolic pathways involved in the accumulation of nutrients. Finally, this study reveals the possible divergence of oil and protein accumulation mechanisms in soybeans.

## MATERIALS AND METHODS

### Plant materials

Soybean plants of the cultivars A7 (Yudou 12) and A35 (Fendou 53) were grown and total RNA was extracted from seeds as described in our previous research (*Zhao et al., 2018b*). For each stage, three independent RNA extractions were performed and used in RNA-seq and qRT-PCR experiments.

### Determination of protein and oil contents of seeds

Determination of the oil content in soybean: the mature seeds were ground to powder which was transferred into 10 mL glass tubes. Oil was extracted with ligarine and total lipids (TL) were determined (*Dong et al., 2011*). The oil content of the soybeans was calculated on the basis of dry weight of the seeds. The content of proteins was determined by Kjeldahl's method. One gram of fine seed powder was used for digestion, cooling and distillation-titration. The percentage of bean protein was the total nitrogen percentage multiplied by 6.25. Each experiment was performed three times.

### cDNA library construction and transcriptome sequencing

The total RNA (2 μg) was sent to GENEWIZ (SuZhou, China) for sequencing, assembly and clustering analysis. The total RNA was quantified and qualified using an Agilent 2100 Bioanalyzer (Agilent Technologies, Palo Alto, CA, USA) and a NanoDrop system (Thermo Fisher Scientific Inc., Waltham, MA, USA). One μg total RNA with RIN value above 7 was used for cDNA library preparation. First, double-strand cDNA was synthesized, and then treated to repair both ends and add a dA-tailing, followed by a T-A ligation to add adaptors to both ends. Size selection of adaptor-ligated DNA was then performed using AxyPrep Mag PCR Clean-up (Axygen, Union City, CA, USA), and fragments of ~360 bp (with the approximate insert size of 300 bp) were recovered. Each sample was then amplified by PCR for 11 cycles using P5 and P7 primers, with both primers carrying sequences which can anneal to the flow cell to perform bridge PCR, and with the P7 primer carrying a six-base index allowing for multiplexing. The PCR products were cleaned up using AxyPrep Mag PCR Clean-up (Axygen, Union City, CA, USA),

validated using an Agilent 2100 Bioanalyzer (Agilent Technologies, Palo Alto, CA, USA), and quantified by Qubit 2.0 Fluorometer (Invitrogen, Carlsbad, CA, USA).

Then libraries with different indices were multiplexed and loaded on an Illumina HiSeq instrument according to manufacturer's instructions (Illumina, San Diego, CA, USA). Sequencing was carried out using a 2 × 150 bp paired-end (PE) configuration. The raw data has been deposited in NCBI with the following accession numbers: A35-2: SRR12283471, SRR13090079, SRR13090078; A35-4: SRR12283470, SRR13090077, SRR13090076; A35-6: SRR12283469; A7-2: SRR12283468, SRR13090082, SRR13090083; A7-4: SRR12283467, SRR13090081, SRR13090080; A7-6: SRR12283466, SRR13100636, SRR13100635.

### RNA-seq data analysis

The software of Trimmomatic v0.30 (http://usadellab.org/cms/index.php?page=trimmomatic) was used to remove the technical sequences for the high quality clean data (*Bolger, Lohse & Usadel, 2014*). As for mapping, the reference genome sequences were downloaded from Phytozome (https://phytozome-next.jgi.doe.gov/info/Gmax_Wm82_a2_v1, version Glyma 2.0, 975 Mb), and then the clean data were aligned to the reference genome using Hisat2 (v2.0.1) (http://daehwankimlab.github.io/hisat2/). Differential expression analysis used the DESeq Bioconductor package, a model based on the negative binomial distribution (http://www.bioconductor.org/packages/release/bioc/html/DESeq2.html) (*Anders & Huber, 2010*, *2012*). After controlling for the false discovery rate (*Benjamini & Hochberg, 1995*), the $P$-value threshold of <0.05 was used for detecting significant differential gene expression. Gene Ontology (GO) and Kyoto Encyclopedia of Genes and Genomes (KEGG) analyses were performed according to the methods described in *Xin et al. (2019)*.

### Principal component analysis

Principal component analysis was performed to visualize sample-to-sample distances, as described by *Lee et al. (2014)*. This analysis was conducted in R (*R Core Team, 2017*).

### Quantitative real time-PCR analysis and statistical tests

The methods for Quantitative Real Time-PCR (qRT-PCR) analysis and statistical tests were performed as described previously (*Zhao et al., 2018b*). The specific primers are shown in Table S1.

## RESULTS

### Phenotype identification and transcriptome sequencing

A7 and A35 are two cultivars of soybean from China. Their oil and protein contents in mature seeds vary greatly, with the average protein content of A7 (50.8%) being significantly higher than in A35 (38.1%), while the oil content in A7 (17.6%) is much lower than in A35 (22.11%) (Fig. 1B). To study the global changes in gene expression during seed development, seeds from both cultivars were collected and analyzed by transcriptome sequencing at different timepoints (weeks after flowering or WAF) resulting in the

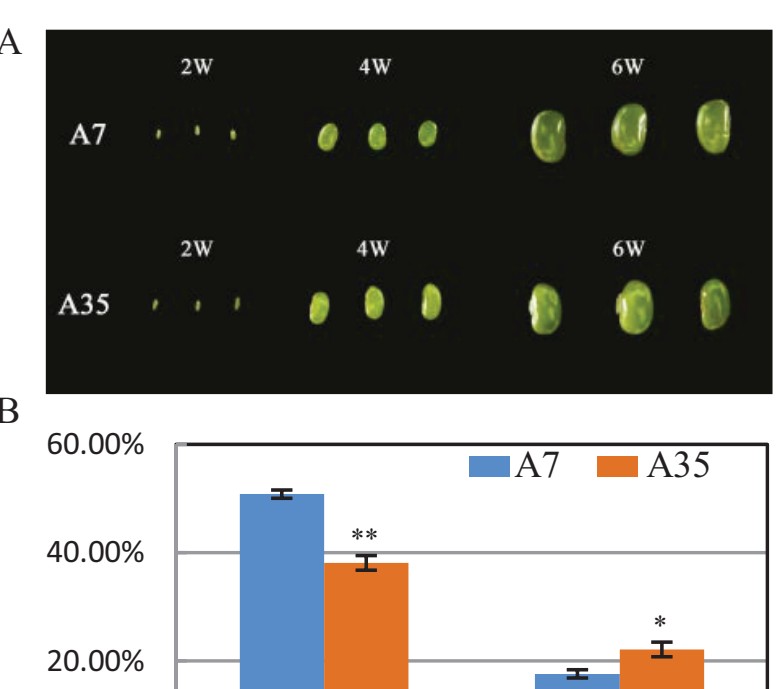

**Figure 1 Seed phenotypes, and the protein and oil contents in A7 and A35.** (A) Phenotypes of two-WAF, four-WAF and six-WAF seeds. (B) The total protein and oil content in the mature seeds of A7 and A35. *X*-axis represented the types of protein and oil. *Y*-axis represented the dry seeds oil and protein percentage. Error bar: standard deviation. The significance was tested in comparison with the contents of oil and protein in A7 (blue columns). The * and ** represent the significance at a $P < 0.05$ and $P < 0.01$ levels, respectively.

following samples: A7-2 WAF, A7-4 WAF, A7-6 WAF, A35-2 WAF, A35-4 WAF, A35-6 WAF (Fig. 1A).

All the sequencing data are shown in Table 1. A total of 139 and 153 million clean reads were acquired in A7 and A35, respectively. The reads were highly matched to the soybean reference genome, and the statistical results showed that 54,665,387, 44,537,617, 40,763,561, 39,120,215, 44,375,847 and 41,099,599 reads were mapped for A7-2, A7-4, A7-6, A35-2, A35-4 and A35-6, respectively, with 90.3% average matching rate. The saturation and evenness of the transcriptomes were further analyzed to estimate the sequencing depth and evenness. According to the gene expression levels, the saturation curves were analyzed in four classifications: below 25%, 25–50%, 50–75% and 75–100% (Fig. S1). The expression evenness was analyzed through the whole transcripts from 5′ to 3′ (Fig. S2). The results showed that the overall quality of sequencing in this study was high and covered the vast majority of expressed genes. In addition, the correlation analysis of different samples showed that the average correlation of all the comparison pairs was more than 90% (Fig. S3).

**Table 1 Summary of soybeans seeds transcriptome data sequenced by the Illumina platform.**

| Sample | A35-2WAF | A35-4WAF | A35-6WAF | A7-2WAF | A7-4WAF | A7-6WAF | Sum/Ave |
|---|---|---|---|---|---|---|---|
| Raw reads | 45,623,515 | 48,593,831 | 46,469,858 | 60,380,729 | 49,145,343 | 44,862,351 | 295,075,627 |
| Raw bases | 6,843,527,200 | 7,289,074,700 | 6,970,478,700 | 9,057,109,400 | 7,371,801,500 | 6,729,352,700 | 44,261,344,200 |
| Q20 (%) | 95.97 | 96.89 | 96.77 | 96.25 | 96.13 | 96.46 | 96.41 |
| Q30 (%) | 90.78 | 92.57 | 92.32 | 91.06 | 90.98 | 91.70 | 91.57 |
| Clean reads | 45,229,794 | 48,357,101 | 46,210,171 | 59,926,314 | 48,762,965 | 44,562,063 | 293,048,408 |
| Clean bases | 6,703,514,302 | 7,186,695,710 | 6,868,762,709 | 8,891,051,948 | 7,243,899,179 | 6,621,751,911 | 43,515,675,759 |
| Q20 (%) | 96.42 | 97.20 | 97.13 | 96.78 | 96.59 | 96.83 | 96.83 |
| Q30 (%) | 91.40 | 92.99 | 92.81 | 91.80 | 91.62 | 92.20 | 92.14 |
| Mapped reads | 39,120,215 | 44,375,847 | 41,099,599 | 54,665,387 | 44,537,617 | 40,763,561 | 264,562,226 |
| Proportion (%) | 86.64 | 91.56 | 89.61 | 91.17 | 91.30 | 91.50 | 90.30 |

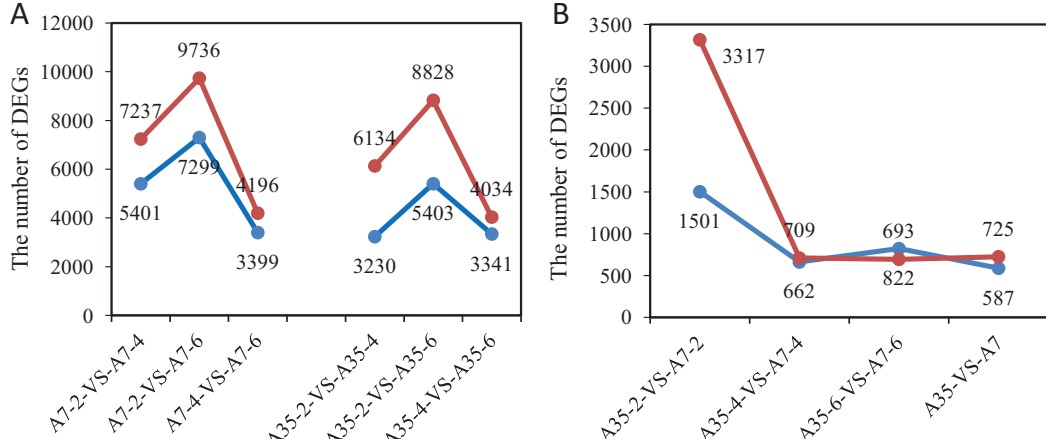

**Figure 2 The gene numbers of significantly different expression among samples.** The *x*-axis represented the comparison pairs. The *y*-axis represented the number of DEGs. (A) The comparison between two-, four-, six-WAF in cultivars. (B) The comparison between A7 and A35. The red and blue lines represent up-regulated and down-regulated DEGs, respectively.

## DEGs in paired comparisons of A7 and A35

In order to detect significant DEGs, the *P*-value was adjusted as above, and the log2 foldchange > 2 and *P*-value < 0.05 were set as thresholds (Fig. S4). In total, the DEGs from different developmental stages and different soybean cultivars have been systematically compared and valued. Comparing different time points (WAF) in the cultivar A7, 12,638 DEGs were found in A7-2-VS-A7-4 (7,237 up-regulated and 5,401 down-regulated); 7,595 DEGs were found in A7-4-VS-A7-6 (4,196 up-regulated and 3,399 down-regulated); and 17,035 DEGs were found in A7-2-VS-A7-6 (9,736 up-regulated and 7,299 down-regulated) (Fig. 2A). Similarly, in A35 the number of DEGs in A35-2-VS-A35-4 was 9,364 (6,134 up-regulated and 3,230 down-regulated); in A35-4-VS-A35-6 it was 7,375 DEGs (4,034 up-regulated and 3,341 down-regulated); in A35-2-VS-A35-6, it was 14,231

DEGs (8,828 up-regulated and 5,403 down-regulated). Generally, the DEGs in A7 and A35 mainly occurred in the 2- and 4- WAF pair and in the 2- and 6-WAF pair, and many more up-regulated genes were observed than down-regulated genes, especially in the early developmental stages (Fig. 2). In seeds, the storage nutrients gradually accumulated from the early stage of the seed development (two WAF) to the late stage of seed development (six WAF). On this basis, we conclude that a large number of genes are activated mainly around two WAF to meet the needs of energy and materials during the accumulation of storage nutrients in seeds.

Furthermore, the comparisons between A7 and A35 at the same stages were also performed. Overall, 1312 DEGs (725 up and 587 down) were found in both A7 and A35 (Fig. 2B). However, at different developmental stages a divergence of the DEGs was observed. Firstly, comparing A7 and A35, the DEGs mainly occurred in 2-WAF seeds, with the number of DEGs in 2-WAF seeds (4,818) being much higher than in the 4- (1,371) and 6-WAF seeds (1,515). This observation was supported by the results of the principal component analysis (PCA) showing a major difference between A7 and A35 at the early stage of seed development (two-WAF seeds) (Fig. S5). Secondly, in the 2-WAF seeds, the number of upregulated DEGs (3,317) was significantly higher than the number of downregulated genes (1,501) in A35 compared to A7, which indicated more genes were activated in A35 than A7 during the early developmental stages of seeds. In addition, the time point specific DEGs were also compared, and the results were consistent with above ones, in which 3,806, 446 and 708 specific DEGs have been identified at two WAF, four WAF and six WAF, respectively.

## Analysis of DEG clustering and pathway enrichment

To better understand the trends of DEG regulation during the seed development, hierarchical clustering was performed and presented in a boxplot dendrogram (Fig. S6). The dendrogram provided a clear overview of the clade structure, in which the DEGs clustered to the up-regulated or down-regulated clades, according to their expression trends. In the heatmap (Fig. S6), 25,139 DEGs were shown, but only half of them (13,262) were annotated in GO involving biological process (5,631), cellular component (1,200) and molecular function (6,431). Furthermore, the DEGs concentrated in the functional groups such as catalytic activity, binding, metabolic process, cellular process and biological regulation, which were consistent with the processes of seed development including energy metabolism, nutrients accumulation and cell proliferation (Fig. 3A; Fig. S7).

To further find out the involved metabolic pathways of the DEGs, the data were mapped against the KEGG pathway database (Table S5). In A7, a total of 85 pathways with $Q$ value < 0.05 were identified. Fifty eight, 46 and 33 pathways were mapped in the A7-2-VS-A7-4, A7-2-VS-A7-6 and A7-4-VS-A7-6 paired comparisons, respectively. In A35, 78 pathways were totally identified: 46 pathways were in A35-2-VS-A35-4, 37 in A35-2-VS-A35-6 and 28 in A35-4-VS-A35-6, respectively. Notably, eight and four significantly mapped pathways were common to the three paired comparisons in A7 and A35, respectively (Fig. 3B). In A7, the common pathways included ko05203 (Viral carcinogenesis), ko01524 (Platinum drug resistance), ko00073 (Cutin, suberine

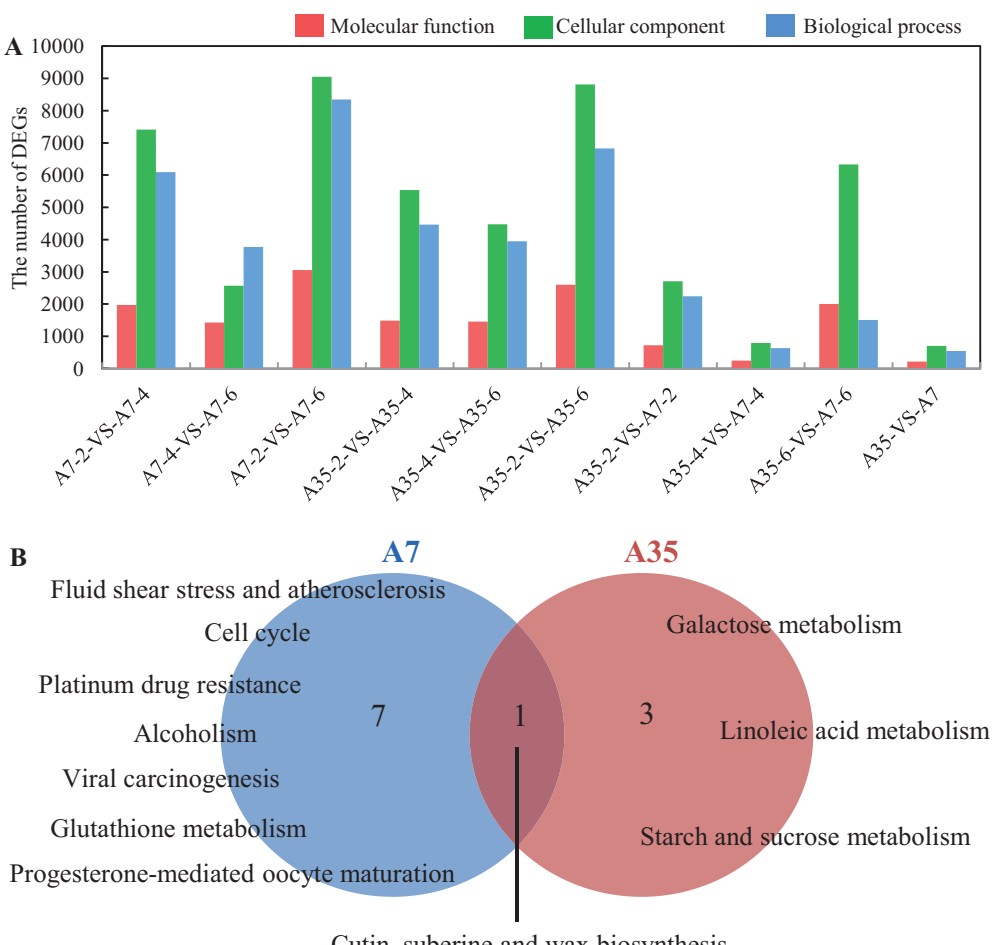

**Figure 3 GO analysis and Pathways enrichment of DEGs in three paired comparisons in A7 and A35.**
(A) The DEGs number distribution of different functional groups in paired comparisons. (B) The common pathways in the three paired comparison in A7 and A35, respectively. The functions related to molecular function, cellular component and biological process were shown in red, green and blue, respectively. The numbers of significantly rich pathways were marked on the map, respectively.

and wax biosynthesis), ko05418 (Fluid shear stress and atherosclerosis), ko00480 (Glutathione metabolism), ko05034 (Alcoholism), ko04110 (Cell cycle) and ko04914 (Progesterone-mediated oocyte maturation), which were mainly involved in cell cycle and stresses responses. However, in A35, the common pathways were ko00052 (Galactose metabolism), ko00073 (Cutin, suberine and wax biosynthesis), ko00591 (Linoleic acid metabolism) and ko00500 (Starch and sucrose metabolism), mainly involved in sugar and fatty acids metabolism. Only one common pathway, namely ko00073, occurred in both A7 and A35, which was related to the biosynthesis of cutin, suberine and wax in cell wall (Fig. 3). When specific developmental stages of A7 were compared with A35, lipid-related pathways (ko00592, ko00062 and ko00071) and sugar-related pathways (ko00196, ko00195, ko00710 and ko00010) were found at the 2-WAF stage, while amino acid metabolism pathways (ko00360 and ko00270) were

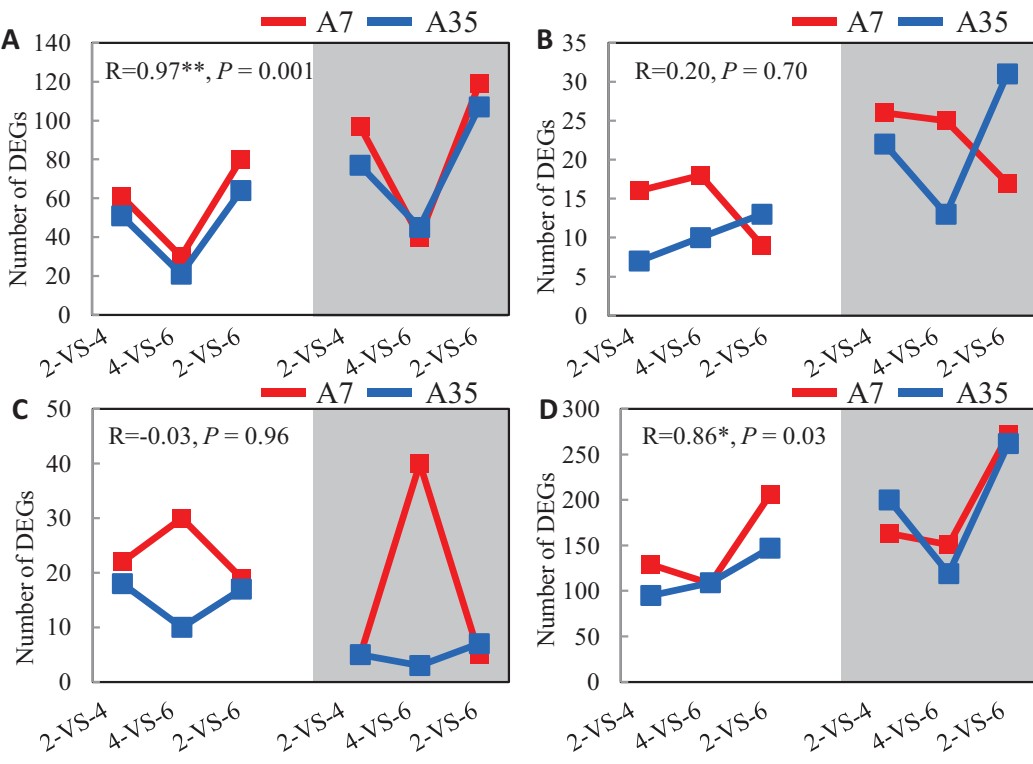

**Figure 4 The analysis of the DEGs numbers involved in oil and protein metabolism.** (A–D) The numbers of DEGs in lipid, fatty acids, proteins and amino acids metabolism in A7 and A35, respectively. The x-axis represented the comparison pairs. The y-axis represented the number of DEGs. The white and gray backgrounds represented the down- and up-regulated DEGs, respectively.

found at 4- and 6-WAF stages, respectively (Table S5). The divergence of the pathways in different developmental stages of soybeans showed conformity with the differences in nutrient contents between A7 and A35.

## Exploration of DEGs that may contribute to the oil and SSP contents in soybeans

Given the significant difference in oil and protein contents in A7 and A35, the DEGs involved in the synthetic and metabolic pathways of lipid, fatty acids, amino acids and proteins during the seed development were specifically investigated (Fig. 4). Firstly, the change trends of up- and down-regulated DEGs involved in the lipid metabolism were V-shaped in A7 and A35 from 2-VS-4, 4-VS-6 to 2-VS-6. Correlation analysis showed that the trends were consistent between A7 and A35 (Fig. 4A). Similarly, the change trends of DEGs related to the amino acid metabolism were also V-shaped in A7 and A35 (Fig. 4D). On the other hand, the change trends of DEGs involved in fatty acid and protein metabolism were different in A7 and A35. As for the fatty acids, the numbers of up- and down-regulated DEGs in A7 were similar in 2-VS-4 and 4-VS-6 pairs, but sharply decreased in 2-VS-6. However, in A35 the number of down-regulated DEGs was increased linearly from 2-VS-4, 4-VS-6 to 2-VS-6, while the trend of the up-regulated DEGs was

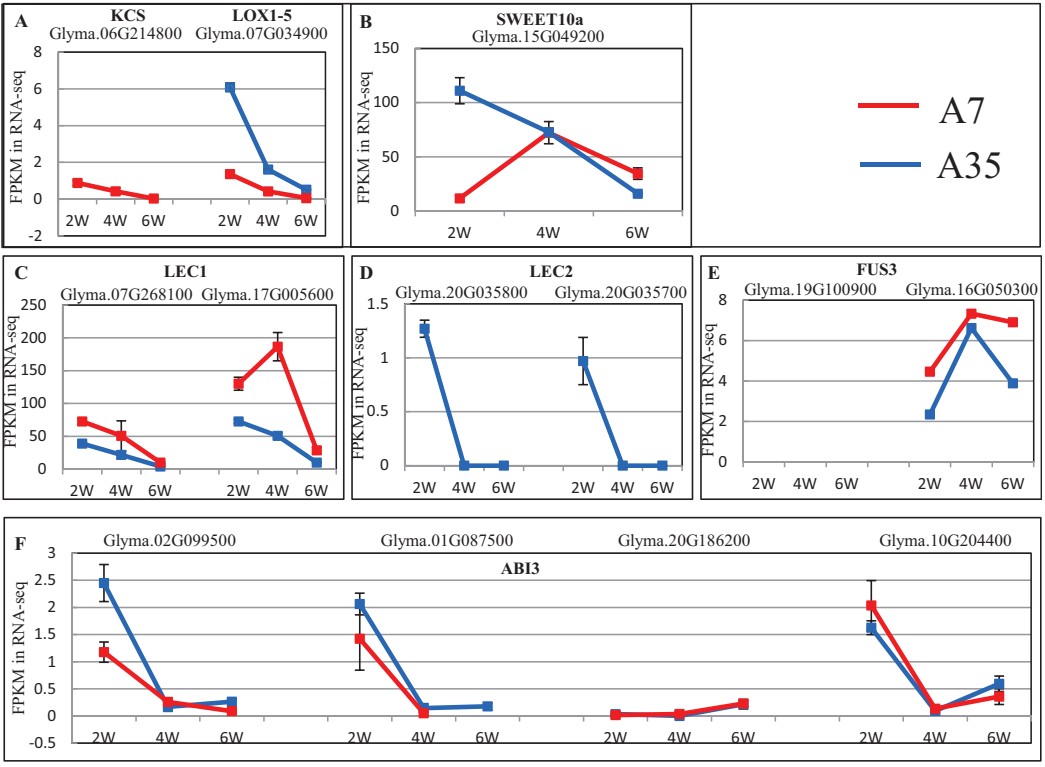

**Figure 5 The expression trends of DEGs related to oil and SSPs.** (A–F) Represent the significant DEGs involving in oil and linoleic acid contents (A), sugar transporter (B), and transcription factors of SSPs expression (C–F), respectively. *X*-axis represented the developmental stages; *Y*-axis represented FPKM in RNA-Seq.

V-shaped (Fig. 4B). Notably, the trends of DEGs involved in protein synthesis were opposite in A7 and A35. In A7, the number of up- and down-regulated DEGs was first increased and then decreased, while in A35 it was the other way around (Fig. 4C). Notably, the divergent trends of numbers of DEGs involved in fatty acid and protein synthesis were consistent with the difference in oil and protein contents in A7 and A35, which indicated that these DEGs probably contribute to the content differences in A7 and A35.

To further reveal DEGs involved in oil and protein content differences in A7 and A35, the significant DEGs were matched to the QTL database in SoyBase (https://www.soybase. org/). In this way, two DEGs, Glyma.06G214800 and Glyma.07G034900, were found to be involved in oil and linoleic acid contents. Glyma.06G214800 encodes 3-ketoacyl-CoA synthase (KCS) catalyzing the elongation of C18 fatty acids while Glyma.07G034900 encodes linoleate 9S-lipoxygenase (LOX1-5) which generates a peroxide by adding oxygen in the double bond of linoleic acid. In our study, the expression levels of Glyma.06G214800 and Glyma.07G034900 were down-regulated during the seed development in A35. However, the expression of Glyma.06G214800 was not detected in A7, while the expression levels of Glyma.07G034900 in A7 were significantly higher than in A35, which probably is related to the difference in oil content between A7 and A35 (Fig. 5A; Table S6).

No specific DEGs involved in protein content were matched to the QTL database. However, previous studies have revealed that the expression of SSPs is mainly controlled by transcription factors during the seed filling stages, the master regulators include LEC1, Leafy Cotyledon 2 (LEC2), Abscisic *acid-intensive 3* (ABI3) and FUSCA3 (FUS3) (reviewed by *Fedorova et al., 2002*). In our study, the expression of *LEC1*s, *LEC2*s, *FUS3*s and *ABI3*s showed some divergence, in which the expression trends of *LEC1*s (Glyma.07G268100 and Glyma.17G005600) and *ABI3* (Glyma.02G099500, Glyma.01G087500, and Glyma.10G204400) were basically consistent in A7 and A35, while *LEC2*s (Glyma.20G035800 and Glyma.20G035700) and *ABI3* (Glyma.20G186200) expression was different (Fig. 5; Table S6). Most of the consistently expressed transcription factors (TF) were higher in A35 than A7 except for *ABI3* (Glyma.02G099500 and Glyma.01G087500) (Fig. 5; Table S6). Considering the positive relationship of TF expression with the content of SSPs, the two DEGs, *ABI3*s (Glyma.02G099500 and Glyma.01G087500) and *LEC2*s (Glyma.20G035800 and Glyma.20G035700), might be correlated with the difference of SSPs between A7 and A35. Furthermore, a sugar transporter, SWEET10a (Glyma.15G049200), contributing to the increase in seed weight and oil content and reduced protein content was identified by the QTL database search. Its expression patterns were different in A7 and A35, with an inverted V shape in A7 and linear decline in A35. The biggest divergence between A7 and A35 occurred at the 2-WAF stage, where the expression in A35 is much higher than in A7 ($p < 0.001$, Fig. 5; Table S6).

## Expression flux in the Fatty acids synthesis pathway

The pathways of fatty acid metabolism mainly include fatty acid biosynthesis (ko00061), fatty acid elongation (ko00062) and fatty acid degradation (ko00071) (Fig. 6; Fig. S8). All the DEGs and their expression trends in these pathways have been marked in Fig. 6 to identify their expression flux. The highest expression levels in the whole pathway were observed for fatty acid dehydrogenase 2 (FAD2s) encoded by Glyma.10G278000 and Glyma.20G111000. Both DEGs were specifically expressed in seeds and the expression levels were increased during the seed development. The expression levels of *FAD2*s were higher in A35-2WAF than in A7-2WAF, which suggested that *FAD2* genes might be essential for the content of oil in A7 and A35. Notably, the expression levels of key enzymes of the fatty acid elongation pathway were drastically reduced during the development of seeds in A7 and A35 (Fig. 6) and their expression levels in A35 were lower than in A7. The results might explain why the fatty acids in soybean seeds were mainly present in the form of C18. As for fatty acid degradation, the expression of enoyl-CoA hydratase/3-hydroxyacyl-CoA dehydrogenase (MFP2/HAD) and acetyl-CoA C-acetyltransferase (atoB) was stable in the development of seeds in A7 and A35 (Fig. 6).

## The qRT-PCR analysis of DEGs in soybeans

To confirm the RNA-seq results in our research, we selected 6 significant DEGs related to the fatty acid metabolism and the synthesis of amino acids for qRT-PCR analysis (Fig. 7; Table S6). In all, the expression results of qRT-PCR were consistent with the RNA-seq results. Among them, the expression of Glyma.13G035200 was significantly increased along

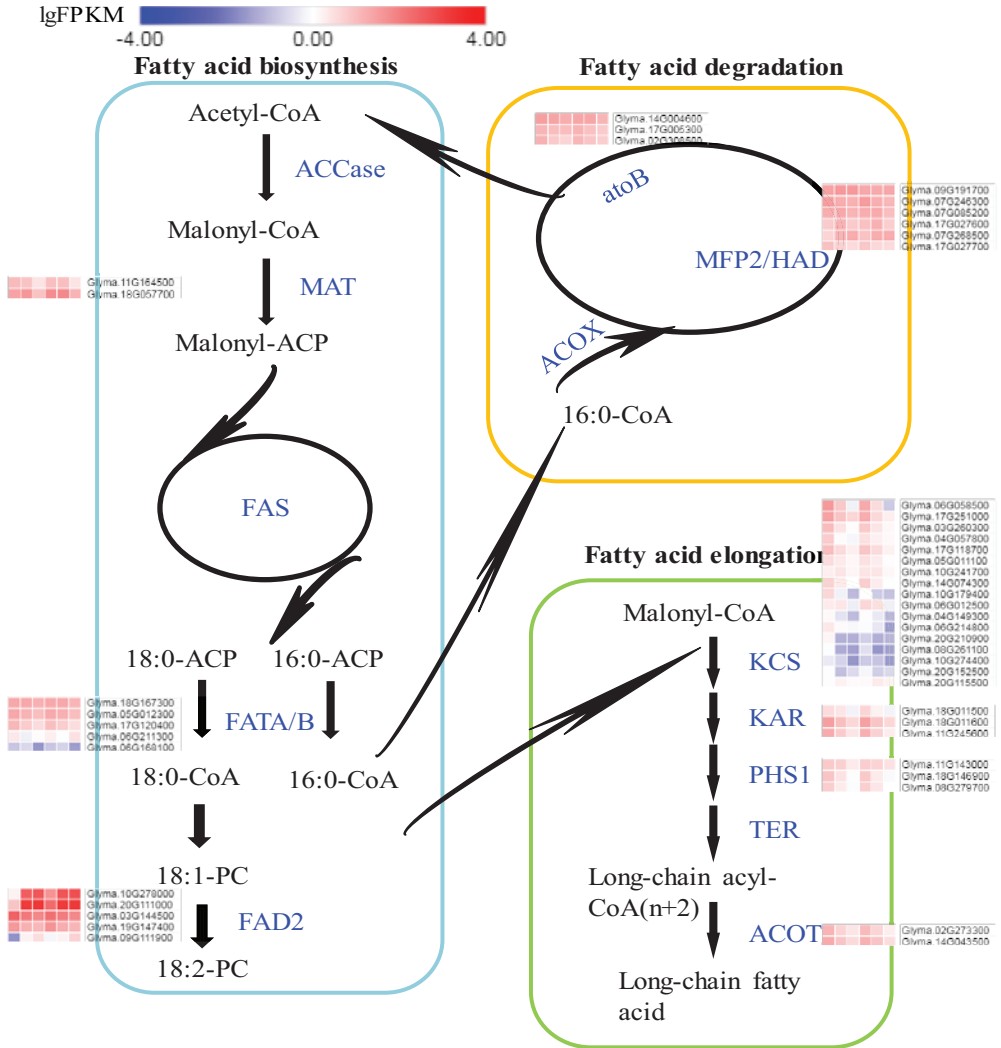

**Figure 6 DEGs in soybeans associated with the fatty acid biosynthetic, elongation and degradation pathways.** Each square represents a comparative pair. The squares from left to right represent: A7-2, A7-4, A7-6, A35-2, A35-4 and A35-6. The red, blue squares indicate their expression levels. The enzymes are marked in blue in pathways. ACCase: acetyl-CoA carboxylase; MAT: malonyl-CoA; ACP transacylase; ACP: acyl carrier protein; FAS: fatty acid synthases; FATA/B: oleoyl-(acyl-carrier-protein) hydrolase; FAD2: fatty acid desaturase 2; ACOX: acyl-CoA oxidase; MFP2: enoyl-CoA hydratase; HAD: 3-hydroxyacyl-CoA dehydrogenase; atoB: acetyl-CoA C-acetyltransferase; KCS: 3-ketoacyl-CoA synthase; KAR: very-long-chain 3-oxoacyl-CoA reductase; PHS1: very-long-chain (3R)-3-hydroxyacyl-CoA dehydratase; TER: very-long-chain enoyl-CoA reductase; ACOT: acyl-CoA thioesterase.

the seed development, while Glyma.16G147300 was gradually decreased. The expression trend of Glyma.06G211300 was decreased from 2-WAF to 4-WAF seeds, and then was increased in 6-WAF seeds. The expression trends of Glyma.17G047000, Glyma.17G027600 and Glyma.06G183900, on the other hand, were bell-shaped, increasing from 2-WAF to 4-WAF seeds and then decreased in 6-WAF seeds. The results verified that our RNA-seq data were reliable.

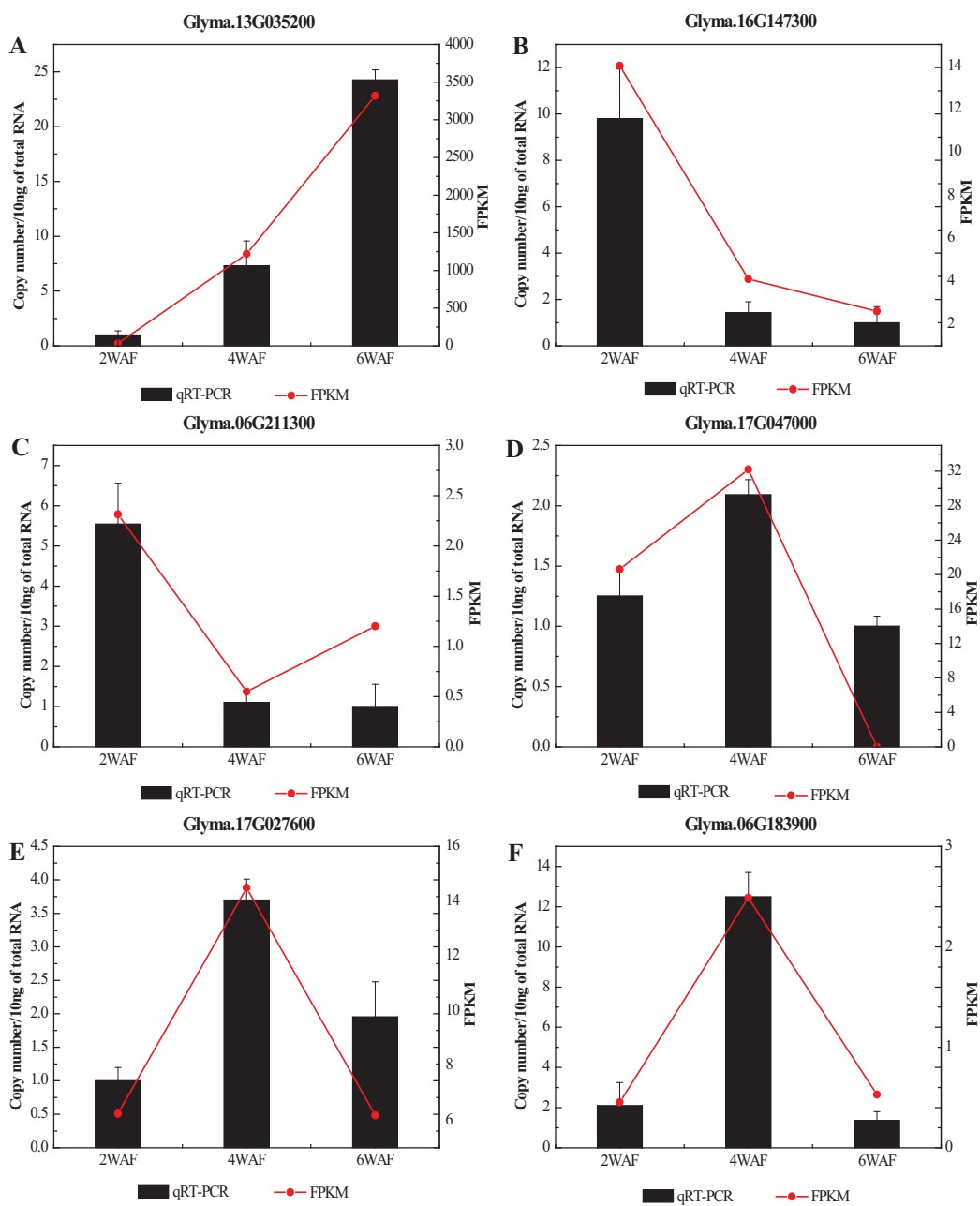

**Figure 7 qRT-PCR analysis of DEGs with different samples from that in RNAseq.** (A–F) Represent the significant DEGs in the metabolism of fatty acids (A, C–E) and amino acids (B, F), respectively. *X*-axis represented the developmental stages, the black columns represented qRT-PCR results, and the red dots represented RNA-Seq results; *Y*-axis represented the relative level of gene expression in qRT-PCR and FPKM in RNA-Seq.

## DISCUSSION

It has been shown that seeds of different soybean cultivars vary widely in nutrient content with a negative correlation between oil and protein contents (*Chung et al., 2003*). In our study, the two soybean cultivars A7 and A35, with similar genetical background, showed distinctive differences in storage protein and oil contents, providing a useful

system for comparative studies of the regulation of seed nutrient accumulation. Our transcriptome data of three stages of seed development, two-, four- and six-WAF seeds, were used to analyze and compare gene expression trends. Similar to previous studies (Bao & Ohlrogge, 1999), there were more DEGs at the early stage of seed development with up-regulated DEGs more numerous than down-regulated ones in both soybean cultivars (Fig. 2A). The accumulation of nutrients, especially oils and SSPs, is essential to seed development. The accumulation of nutrients is a complex and dynamic process, which is influenced by multiple genetic and environmental factors (Gupta et al., 2017; Hills, 2004). Until now, studies including genetics, omics, QTL and GWAS have been performed to investigate the mechanisms of seed filling (Hyten et al., 2004a, 2004b; Li et al., 2018a, 2018b; Gupta et al., 2017; Agrawal et al., 2008). It has been revealed that the oil content of soybean seeds gradually increased until 40 days after flowering (DAF), and stayed steady in later stages (Li et al., 2015; Collakova et al., 2013), which is consistent with our results. In our study, the comparison of transcriptome data from different developmental stages of seeds and different soybean cultivars revealed that the change trends of DEGs involved in the synthesis of fatty acids were basically consistent in A7 and A35, including the down-regulation of fatty acid synthesis in 6-WAF seeds. It has been reported that the supply of fatty acids was the limiting factor for the accumulation of oil in embryos (Bao & Ohlrogge, 1999). Linoleic acid is the main fatty acid of soybean oil. FAD2 is essential to convert oleic acid into linoleic acid. Notably, FAD2 genes involved in the fatty acid pathways were highly expressed in all periods, and in A35 it was obviously higher than in A7, which indicated that the oil difference in A35 and A7 might be attributed to the expression difference of FAD2 genes. In our previous study, it was also shown that the bioactivity of FAD2 was correlated with oil content in different plants (Zhao et al., 2019). Therefore, the oil content of seeds might be controlled by both the expression and bioactivity of FAD2. Furthermore, the DEGs in the fatty acid degradation pathway were similar in A7 and A35, and the result showed that the degradation pathway was not the main reason for divergence in the two cultivars. Furthermore, we analyzed the fatty acid elongation pathway and found that the expression trends were drastically reduced during the seed development in A7 and A35 (Figs. 5 and 6). Moreover, the expression levels of KCS in A35 were lower than in A7. The results indicated that more fatty acids were present in the form of C18 in the late development of seeds in A35, which was consistent with the composition of soybean oil. Based on the above results, it can be concluded that the differences in gene expression related to biosynthesis and elongation of fatty acids during the seed development contributed to the difference of oil content in A35 and A7. More experiments need to be done for more details.

In addition, the content of proteins was also significantly different in A7 and A35. The change trends of DEGs involved in the proteins synthesis were different, especially in 4-VS-6 of A7 and A35 (Fig. 4C), which suggested the content differences between A7 and A35 might be due to the expression levels of the genes encoding proteins between four WAF seeds and seed WAF seeds. Unfortunately, in our study, we didn't find the identified

protein DEGs which matched into the QTL database. The possible reason might be that protein content is a complex quality trait and is influenced by many different genes. TFs are essential to regulate the gene expression. Previous studies have revealed that the expression of SSPs is mainly controlled by TFs, *LEC1*, *LEC2*, *ABI3* and FUSCA3 (*FUS3*), during the seed filling stages (reviewed by *Verdier & Thompson (2008)*). These TFs interacted in a network in which LEC1 induces the expression of *LEC2*, *ABI3* and *FUS3*. Moreover, *LEC1* and *LEC2* genes regulate each other, and activate the expression of *FUS3* and *ABI3*, and then activate the *SSP* genes (*Fedorova et al., 2002*; *Kagaya et al., 2005*; *Santos Mendoza et al., 2005*; *Keith et al., 1994*; *Parcy et al., 1997*; *To et al., 2006*; *Braybrook et al., 2006*). In our study, the expression of *LEC1*s, *LEC2*s, *FUS3*s and *ABI3*s showed some divergence, in which the expression trends of *LEC1*s (Glyma.07G268100 and Glyma.17G005600), *ABI3* (Glyma.02G099500, Glyma.01G087500 and Glyma.10G204400) were basically consistent in A7 and A35, while *LEC2*s (Glyma.20G035800 and Glyma.20G035700) and *ABI3* (Glyma.20G186200) showed different trends (Fig. 5; Table S6). Besides, most of consistently expressed TFs were higher in A35 than A7 except for *ABI3* (Glyma.02G099500 and Glyma.01G087500) (Fig. 5; Table S6). Considering the positive relationship of TF expression and content of SSPs, we suggest that the two DEGs, *ABI3*s (Glyma.02G099500 and Glyma.01G087500) and *LEC2*s (Glyma.20G035800 and Glyma.20G035700), are correlated with the difference of SSPs between A7 and A35. In addition, an important sugar transporter, SWEET10a, has been identified to be involved in the seed size, and oil and protein contents (*Wang et al., 2020*), and in our study very similar results were observed. Note that our results further indicated that the difference in expression patterns and levels of SWEET10a at the early stage of seed development might be related to the different protein and oil contents of soybean seeds. More research will be needed to confirm this relationship.

It has been reported that the synthesis of special amino acids, such as methionine and asparagine, is also related to the accumulation of proteins (*Li et al., 2015*; *Galili, Amir & Fernie, 2016*; *Molvig et al., 1997*; *Zhao et al., 2018a*, *2018b*). It has also been reported that asparagine was the major form of nitrogen imported into seeds from the vegetative organs. The asparaginase enzyme (Glyma.05G018300) was predicted to interconvert amino acids for protein synthesis during seed filling (*Li et al., 2015*). In our study, the expression of Glyma.05G018300 was also analyzed, but no significant differences were found in any comparison indicating the gene might be not involved in the protein differences in the studied cultivars (Fig. S9).

## CONCLUSIONS

In this study, comparative transcriptome analysis was performed to study the dynamic gene expression at different developmental stages of soybean seeds in the A7 and A35 cultivars. The results showed that more DEGs occurred at early stages of seed development, especially at the two WAF stage, and there were more up-regulated DEGs at the early stages compared with the late stages. The change trends of the DEGs involved in metabolism of nutrients such as oil and SSPs were different. The results indicate that DEGs related to the synthesis and elongation of fatty acids, *FAD2*, *KCS* and *LOX*, contribute to

the oil content difference in A7 and A35. The data also suggest that the accumulation of SSPs is controlled by transcription regulators such as *ABI3* and *LEC2*, while the sugar transporter *SWEET10a* might contribute to both oil and SSP contents.

### Funding

This work was supported by Zhejiang Provincial Major Agriculture Science and Technology Special Sub-project (Grant No. 2016C02050-10-3). The funders had no role in study design, data collection and analysis, decision to publish, or preparation of the manuscript.

### Grant Disclosures

The following grant information was disclosed by the authors:
Agriculture Science and Technology Special Sub-project: 2016C02050-10-3.

### Competing Interests

The authors declare that they have no competing interests.

### Author Contributions

- Li Peng conceived and designed the experiments, prepared figures and/or tables, authored or reviewed drafts of the paper, and approved the final draft.
- Linlin Qian performed the experiments, prepared figures and/or tables, and approved the final draft.
- Meinan Wang performed the experiments, prepared figures and/or tables, and approved the final draft.
- Wei Liu performed the experiments, prepared figures and/or tables, and approved the final draft.
- Xiangting Song performed the experiments, prepared figures and/or tables, and approved the final draft.
- Hao Cheng performed the experiments, prepared figures and/or tables, and approved the final draft.
- Fengjie Yuan conceived and designed the experiments, prepared figures and/or tables, authored or reviewed drafts of the paper, and approved the final draft.
- Man Zhao conceived and designed the experiments, prepared figures and/or tables, authored or reviewed drafts of the paper, and approved the final draft.

### Field Study Permissions

The following information was supplied relating to field study approvals (i.e., approving body and any reference numbers):

Field experiments were approved by the local government, Fuyang, Hangzhou (33010347000441X).

## DNA Deposition

The following information was supplied regarding the deposition of DNA sequences:

The raw data are available at NCBI: SRR12283471 (A35-2), SRR12283470 (A35-4), SRR12283469 (A35-6), SRR12283468 (A7-2), SRR12283467 (A7-4), SRR12283466 (A7-6), SRR13090083 (A7-2b), SRR13090082 (A7-2c), SRR13090081 (A7-4b), SRR13090080 (A7-4c), SRR13100636 (A7-6b), SRR13100635 (A7-6c), SRR13090079 (A35-2b), SRR13090078 (A35-2c), SRR13090077 (A35-4b), SRR13090076 (A35-4c).

## Data Availability

Raw data is available in the Supplemental Files.

## Supplemental Information

Supplemental information for this article can be found online at http://dx.doi.org/10.7717/peerj.10772#supplemental-information.

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
