# Peer review of "Comparative transcriptome analysis during seeds development between two soybean cultivars"

_PeerJ, doi:10.7717/peerj.10772_

## Round 0.1 · original submission · Minor Revisions

Thank you for your article. Three reviewers have read and commented on it and all three agree that the manuscript is clear and acceptable but needs revisions.

Please address all of the reviewers' comments and re-submit a revised copy.

Reviewer 1 ·

Basic reporting

In this paper, Peng et al. reported a differential gene expression analysis between low and high oil soybean cultivars at three time points during seed development. The manuscript is well structured, the documentation of methods is sufficient. The figures and tables are in well shape and readable.

Experimental design

However, my main concern is the lack of replicates for each time points, which makes the differential gene expression analysis less reliable. The whole analysis is very descriptive and failed to identify novel genes/pathways involved in the oil and protein content differences between these two soybean cultivars.

1. What is the phylogenic relationship between these two soybean cultivars? How different are they genetically?
2. Do the author think other than gene expression differences, genetic differences between these two soybean cultivars could contribute to the oil and protein content differences?
3. The authors should try to call SNPs/Indels using RNA-Seq data. It could be a useful resource and may gain new insight in into the phenotypic differences.

Validity of the findings

1. Figure 1b, what does the percentage of Y axis mean in the bar plot? In the main text, the authors state “The total content of proteins and oil was measured in mature seeds”. Is this seed stage included in one of the three timepoints surveyed by RNA-Seq?
2. Based on the method described in the manuscript, it is unclear that whether the determination of oil content was normalized based on seed weight.
3. Figure 2, how do these differentially expressed genes identified overlap with different comparison groups? To be more biological reverent, it is better to show whether time point or sample specific differentially expressed genes can be detected. But duo to the lack of replicates at each time point, the time point comparison results would be less reliable.
4. Figure 3, it is unclear whether all DEGs are used or only time point specific highly expressed genes are used for the GO enrichment analysis.
5. Figure 4, comparing the numbers of DEGs identified could be misleading. Genes up-regulated in one comparison could be down-regulated in another comparison. Just showing the numbers is not sufficient, or biological meaningful. It is better to use volcano plots to show the degrees of differences of expression and highlight known genes involved in the oil content pathways.
6. Figure 5, without replicates at each time point, these comparisons have no statistical power.
7. Figure 7, it seems only DEGs are shown. It should include negative controls. For example, some housekeeping genes.

Reviewer 2 ·

Basic reporting

no comment

Experimental design

no comment

Validity of the findings

no comment

Additional comments

1. The DEGs of the synthesis and elongation of fatty acids, FAD2, KCS and LOX, have been revealed might contribute to the oil content difference in A7 and A35, while SSPs might be due to the transcription regulators such as ABI3 and LEC2.
The authors should provide evidence for molecular detection of the expression levels of these genes in A7 and A35 seeds at different stages of development.
2, Line 103: beans should be soybean.
3, Line 109: cConstruction should be Construction.

Reviewer 3 ·

Basic reporting

This manuscript is clear and readable. It has acceptable literature references and background/context provided.The manuscript has professional article structure, and figures. However, it is not really hypothesis driven.

Experimental design

The experimental design is appropriate for this experiment. There lacks a fundamental research questions besides the mention that the genetic mechanisms for the differences in seed composition has not be revealed. Methods are described with sufficient detail & information to replicate.

Validity of the findings

The authors have done a lot of work on this research yet I fail to see the impact and novelty of this research among the many already published in this particular research question. Data analysis has been provided. Conclusions could be improved. Right now they read as the conclusion for results and not addressed to the research community.

Additional comments

General Comments:
The authors of the study “Comparative transcriptome analysis during seeds development between two soybean cultivars set out to identify the genetic mechanisms for the differences in seed composition between two soybean cultivars with distinct protein and oil content using RNA-Seq technology. Overall, this manuscript addresses a major problem in soybean breeding which is the negative association between protein and oil. It is well written, and the authors have done a lot of work to identify the dynamic expression for the seed development process in soybean by evaluating DEG’s for protein vs oil in soybean.

Specific comments:
Line 48-: I believe your introduction has a lot of background and literature but does not demonstrate how this study justifies the novelty and the gap being filled. I believe there have been other studies that have addressed the genetic mechanisms for the differences in seed composition.
Lines 145: I thank the authors for their detailed explanation for the PCA analysis. Where any R statistical packages used to run the PCA. Please include any additional details if any were used.
I believe the authors of this study have demonstrated a lot of work to understand the genetic mechanisms for the differences in seed composition over time. But I would recommend improving on this study is novel from other studies and can be useful for other soybean researchers or breeders.

---

## Round 0.2 · Minor Revisions

Thank you for your manuscript. At this point, I am willing to accept the manuscript on the condition that you ensure all data have been correctly uploaded and stored with NCBI SRA (see reviewer 1's comments) and that you give it a thorough read for grammatical and spelling errors (e.g. Figure 1 'DREG' should be 'DEG'). Please ensure there are no other such cases throughout the manuscript.

Thank you for your work and again for the manuscript.

Reviewer 1 ·

Basic reporting

no comment

Experimental design

no comment

Validity of the findings

no comment

Additional comments

The authors have addressed nearly all my concerns.

However, in the rebuttal letter, the author mentioned that there are three replicates at each time point. There are three time points compared in the manuscript. The data deposited in the NCBI SRA only have 6 samples (method section). The authors need to address this discrepancy.

Reviewer 2 ·

Basic reporting

The manuscript is well structured, the methods is sufficient. The figures and tables are in well readable. It has acceptable literature references and background.

Experimental design

The experimental design is appropriate for this experiment.

Validity of the findings

The authors have done a lot of work on this research.the results are reliable.

Additional comments

The author has answered my questions, I don't have any other questions. In my opinion, it is suitable for publication.

---

## Round 0.3 · Minor Revisions

Thank you for your resubmission.

I have read your rebuttal letter and am not wholly satisfied with the changes the authors have provided. I have two concerns. My first concern is the authors' response to their data archiving. The authors point out that some data were destroyed (A35-6b and A35-6c). This is regrettable but it is then unclear if that destroyed data was used in the analysis. I feel it is inappropriate to use unavailable data in a paper--the data on NCBI should be that which is used in the analysis. The authors need to clarify this in their paper: did they use the deleted data for analysis? If so, I feel they should repeat the analysis with only the publically available data (and for only conditions with appropriate sample sizes) or upload the raw counts for their analysis as supplemental material. The sampling information, including the sample sizes used in the analysis, should all be very clearly printed in Table 1.

My second concern is that the authors need to again read their manuscript for clarity. The authors stated they read through and made changes to clarify the text and also improve grammar and spelling. I see they have done some of this and I do appreciate their efforts. However, there are many spots that may have slipped through their attention (e.g. 'The change trends of DEGs involved in the expression of proteins were different,' and 'In soybean, there is not only different nutrients content, but also a negative correlation of oil and protein content in seeds').

---

## Round 0.4 · accepted · Accept

Thank you for your effort to improve the manuscript and address the outstanding issues.